# Elective Ascending Aortic Aneurysm Surgery in the Elderly

**DOI:** 10.3390/jcm12052015

**Published:** 2023-03-03

**Authors:** Feyza Memis, Carlijn G. E. Thijssen, Arjen L. Gökalp, Maximiliaan L. Notenboom, Frederike Meccanici, Mohammad Mostafa Mokhles, Roland R. J. van Kimmenade, Kevin M. Veen, Guillaume S. C. Geuzebroek, Jelena Sjatskig, Franciscus J. ter Woorst, Jos A. Bekkers, Johanna J. M. Takkenberg, Jolien W. Roos-Hesselink

**Affiliations:** 1Department of Cardiology, Erasmus University Medical Center, 3015 CN Rotterdam, The Netherlands; 2Department of Cardiology, Radboud University Medical Center, 6525 GA Nijmegen, The Netherlands; 3Department of Cardiothoracic Surgery, Erasmus University Medical Center, 3015 GD Rotterdam, The Netherlands; 4Department of Cardiothoracic Surgery, Utrecht University Medical Center, 3584 CX Utrecht, The Netherlands; 5Department of Cardiothoracic Surgery, Radboud University Medical Center, 6525 GA Nijmegen, The Netherlands; 6Department of Cardiothoracic Surgery, Catharina Hospital, 5623 EJ Eindhoven, The Netherlands

**Keywords:** aortic aneurysm, elderly, aortic surgery, ascending aorta, aortic root

## Abstract

Background. No clear guidelines exist for performing preventive surgery for ascending aortic (AA) aneurysm in elderly patients. This study aims to provide insights by: (1) evaluating patient and procedural characteristics and (2) comparing early outcomes and long-term mortality after surgery between elderly and non-elderly patients. Methods. A multicenter retrospective observational cohort-study was performed. Data was collected on patients who underwent elective AA surgery in three institutions (2006–2017). Clinical presentation, outcomes, and mortality were compared between elderly (≥70 years) and non-elderly patients. Results. In total, 724 non-elderly and 231 elderly patients were operated upon. Elderly patients had larger aortic diameters (57.0 mm (IQR 53–63) vs. 53.0 mm (IQR 49–58), *p* < 0.001) and more cardiovascular risk factors at the time of surgery than non-elderly patients. Elderly females had significantly larger aortic diameters than elderly males (59.5 mm (55–65) vs. 56.0 mm (51–60), *p* < 0.001). Short-term mortality was comparable between elderly and non-elderly patients (3.0% vs. 1.5%, *p* = 0.16). Five-year survival was 93.9% in non-elderly patients and 81.4% in elderly patients (*p* < 0.001), which are both lower than that of the age-matched general Dutch population. Conclusion. This study showed that in elderly patients, a higher threshold exists to undergo surgery, especially in elderly females. Despite these differences, short-term outcomes were comparable between ‘relatively healthy’ elderly and non-elderly patients.

## 1. Introduction

Life expectancy is increasing worldwide. The World Health Organization predicts that between 2015 and 2050, the proportion of the world’s population aged over 60 years will nearly double from 12% to 22% [1]. Due to this demographic shift and the fact that cardiovascular disease is still the leading cause of death [2,3], more elderly patients will become eligible for invasive cardiac and aortic interventions. The transition to an aged society raises uncertainties for cardiologists and cardiothoracic surgeons due to frailty and comorbidities that are more often seen in elderly patients [4] and the fact that increasing age is an important risk factor for postoperative morbidity and mortality [5]. In order to improve patient-specific decision-making, it is important to investigate outcomes after major surgery, specifically in elderly patients. Several research gaps exist when it comes to aortic root and/or ascending aortic (AA) aneurysm surgery in the elderly population.

First, most studies investigating the elderly population do not primarily focus on elective surgery of the aortic root and ascending aorta, with the exclusion of acute surgery such as aortic ruptures and dissections [6,7]. Short-term outcomes differ significantly between acute and elective surgery. Therefore, these studies might not represent the actual outcomes of elective ascending aortic replacements in the elderly. Especially in preventive settings, there is ample time to account for all the baseline characteristics available and carefully discuss with the patient the benefits and risks of intervention, but also of not performing an intervention at all, in a shared decision-making process.

Second, studies report contradictory results on mortality and morbidity in elderly patients after (elective) AA surgery. Some studies conclude that outcomes after an ascending aorta and/or (hemi-)arch reconstruction are acceptable with a short-term mortality rate of between 2.1% and 13.5% in elderly patients [8,9]. On the contrary, other studies conclude that elderly patients showed an operative risk with higher postoperative mortality, morbidity, and prolonged admission in the hospital [7,10]. Most importantly, long-term mortality in elderly patients who underwent elective AA surgery has been investigated in very few studies [7,9]. Therefore, due to this scarce and contradictory scientific evidence, the decision to opt for elective aortic surgery at an advanced age remains difficult.

Third, only a few studies have evaluated the patient and procedural characteristics of AA surgery in elderly patients compared to non-elderly patients. Hardly any distinction is made between the different surgical techniques that were performed. Studies indicate on which part of the aorta surgery is performed, but do not report specific distinctions in surgical techniques [7,9].

Therefore, this multicenter study aims to gain more insight into elective AA surgery in the elderly by evaluating the patient and procedural characteristics and comparing early and long-term outcomes after surgery between elderly and non-elderly patients.

## 2. Methods

### 2.1. Study Design

A retrospective multicenter observational cohort study was performed, covering patients who underwent elective surgery for aneurysmal disease of the aortic root and/or ascending aorta in three experienced centers for aorta surgery in The Netherlands between January 2006 and December 2017. The following institutes participated in this study: Erasmus Medical Center (Rotterdam, The Netherlands), Radboud University Medical Center (Nijmegen, The Netherlands) and Catharina Hospital (Eindhoven, The Netherlands). The study was approved by the local ethics committees (MEC-2018-1535) of all participating centers, and was designed, performed, and controlled in accordance with current local and international good clinical practice guidelines. Patient consent was waived due to the retrospective nature of this study, involving only patient files. All study data was anonymized using a study patient ID.

### 2.2. Study Population

Patients were eligible for inclusion if they underwent elective surgery for aneurysmal disease of the aortic root and/or ascending aorta. Patients had to be 18 years old or older at the time of surgery. Elective surgery was defined as surgery that was planned 14 days or longer in advance. Patients who underwent surgery with concomitant cardiac surgical procedures, such as coronary artery bypass grafting (CABG) and mitral valve surgery, and aneurysms extending into the aortic (hemi-)arch including the descending aorta, were included in the analysis. Exclusion criteria were aortic dissection and/or rupture, other emergency aortic surgery, intramural hematoma, penetrating aortic ulcer, pseudo aneurysm or mycotic aneurysm, isolated reduction aortoplasty without replacement, and aneurysms limited to the descending and/or abdominal aorta. Patients who underwent surgery for infected aortic prosthesis or endocarditis of the aortic valve were also excluded. A description of the surgical procedures used is presented in the Appendix A. Patients were stratified into two groups in the analysis: the elderly patient group aged 70 years or older, and the non-elderly patient group aged under 70 years. 

### 2.3. Data Collection

Patients were identified using the institutional aortic surgery databases. Additionally, a profound search was performed using the hospitals’ diagnosis registration systems. Files of all patients with diagnosis treatment codes (DBC’s) related to any aortic disease were checked manually to see if patients were eligible for inclusion. Data were collected from patient files using standardized case report forms and documented in an online clinical data management system, OpenClinica (OpenClinica, LLC, Version: 3.12.2). Medical history, clinical presentation including perioperative data, laboratory results, and post-operative (in-hospital) outcomes were extracted from the hospital’s electronic medical records. Short-term postoperative mortality and morbidity was defined as an event occurring during hospital admission or within 30 days of surgery. Long-term survival data were obtained from Dutch municipal registries. In addition to the absolute aortic diameter, the indexed aortic diameter was calculated by dividing the absolute aortic diameter by Body Surface Area (BSA), which was calculated using the Du Bois formula [11]. Valve-related postoperative complications were defined according to the 2008 Akins Guidelines for Reporting Mortality and Morbidity after Cardiac Valve Interventions [12]. All variables and definitions are shown in the Appendix A.

### 2.4. Statistical Analysis

Statistical analysis was performed using IBM SPSS Statistics, version 24.0 (IBM Corp., Armonk, NY, USA) and computing program R (R Foundation for Statistical Computing, Vienna, Austria. Version 3.6.1). Continuous variables were presented as mean ± standard deviation (SD) when normally distributed, and as median (interquartile range, IQR) when skewed. Categorical data were presented as frequencies (percentages). A comparison was made between the elderly patient group aged 70 years or older and the non-elderly patient group aged under 70 years. Student’s t-test was used to compare continuous variables with normal distributions and the Mann–Whitney U test was used for variables without normal distributions. Furthermore, Pearson’s Chi-squared or Fisher’s exact test was used to compare categorical variables. A subgroup analysis for patients 75 years and older was performed as well, which can be found in the Appendix A. The Kaplan–Meier survival estimation was used to analyze the postoperative survival rate. The difference in survival probability between elderly and non-elderly patients was calculated using the Log-rank test. Data from the Dutch Central Bureau of Population Statistics (CBS) was used to provide a visual comparison between survival in the elderly and non-elderly groups with respect to the general Dutch population. Missing data was handled via multiple imputation with five iterations. Only variables with less than 15% missing were eligible for imputation. Missing data patterns were studied in order to identify and exclude variables with data missing not at random. For imputation, the monotone method was used if the data showed a monotone pattern of missing values, otherwise, fully conditional specification was used. Cox regression analysis using the backwards selection method was performed in order to identify determinants associated with long-term mortality. Only patients who survived at least 30 days after surgery were included in this long-term analysis. Cox regression analysis was performed for the total population and stratified for elderly and non-elderly patients. Based on univariable analysis (*p*-value < 0.05) and clinical relevance, variables were selected for multivariable analysis. The tests were considered statistically significant if the *p*-value was less than 0.05.

## 3. Results

### 3.1. Patient Characteristics

A total of 955 patients were included: 231 patients (24.2%) were stratified in the elderly group and 724 patients (75.8%) in the non-elderly group. Baseline characteristics are presented in Table 1. The median age of the elderly was 74 years (IQR 72–77) and that of the non-elderly was 55 years (IQR 45–63). The elderly were more often female (51.7% vs. 29.7%, *p* < 0.001) and had more cardiovascular risk factors compared to the non-elderly group (Table 1). Moreover, elderly patients had a significantly higher maximal absolute (57.0 mm (IQR 53–63) vs. 53.0 mm (IQR 49–58)) and BSA-indexed (30.3 mm (IQR 26.9–34.5) vs. 25.9 mm (IQR 23.4–29.2)) aortic diameter at the time of AA surgery compared to non-elderly patients. When comparing elderly male and female patients, elderly females had a larger maximal absolute aortic diameter compared to elderly males with a mean of 61.2 ± 9.7 mm vs. 56.9 ± 8.5 mm (with a median of 59.5 mm (IQR 55–65) vs. 56.0 (51–60), *p* < 0.001). In non-elderly patients, females had a mean aortic diameter of 53.2 ± 8.2 mm vs. 54.1 ± 7.9 in males (median 52.0 mm (IQR 48–58) vs. 53.0 (49–58), *p* = 0.085). Cardiovascular risk factors, as mentioned in Table 1, were compared as well between elderly males and females. Elderly males more often had a history of prior cardiac surgery (11.6% vs. 2.5%, *p* = 0.007) and of prior aortic surgery (8.9% vs. 1.7%, *p* = 0.014). Furthermore, elderly males more often had a bicuspid aortic valve (BAV) compared to elderly females (26.8% vs. 10.9%, *p* = 0.002). No disproportionate differences were found in the elderly and non-elderly patients included from each of the different centers.

### 3.2. Operative Characteristics

Table 2 shows the operative characteristics of elderly versus non-elderly patients. Supracoronary aorta replacement (SCAR) was more common in the elderly group (54.3% vs. 36.5%, *p* < 0.001). The elderly more often received biological prostheses, whereas the non-elderly received more mechanical prostheses (both Bentall and aortic valve replacement [AVR]). The David technique was performed less often in the elderly (3.0% vs. 15.2%, *p* < 0.001). Furthermore, elderly patients received significantly more concomitant procedures during AA surgery (66.8% vs. 49.6%, *p* < 0.001).

### 3.3. Short-Term Postoperative Outcomes

Outcomes after AA aneurysm surgery showed that in-hospital or 30-day mortality after elective aortic aneurysm surgery was 1.9% (*n* = 18), with no significant difference between the study groups (3.0% vs. 1.5%, *p* = 0.16; Table 3). Even when the age limit was 75 years, there was no significant difference in mortality (3.8% vs. 1.6%, *p* = 0.13; Appendix A). In-hospital or 30-day mortality was 3.6% in elderly males (*n* = 4) and 2.5% in elderly females (*n* = 3), which was not significantly different (*p* = 0.642). Besides, no significant differences existed in mortality between the elderly and non-elderly in the three centers.

Table 3 shows short-term outcomes after surgery in elderly compared to non-elderly patients. Prolonged hospital admission (more than 20 days) was significantly more common in the elderly (11.3% vs. 6.2%, *p* = 0.044) and more reoperations were performed on elderly patients (32.3% vs. 21.4%, *p* < 0.001).

Short-term postoperative morbidity outcomes are shown in Table 4. Postoperatively, the elderly were significantly more often diagnosed with new supraventricular arrhythmias (39.4% vs. 21.3%, *p* < 0.001), delirium (28.6% vs. 11.0%, *p* < 0.001), infections which were mainly pneumonia (9.5% vs. 5.2%, *p* = 0.02), and severe wound infections (3.0% vs. 0.4%, *p* = 0.003). When comparing elderly female to elderly male patients, there were no significant differences in duration of hospital admission, duration of ICU stay, duration of ventilator support, or number of patients who needed a reoperation within 30 days.

Before AA surgery, estimated Glomerular Filtration Rate (eGFR) levels were significantly lower in elderly patients, as shown in Figure 1 (eGFR 67 mL/min/1.73 m^2^ vs. 86 mL/min/1.73 m^2^*, p* < 0.001). Decreases in eGFR were observed in both groups after surgery. The average decrease was 16.2% (average eGFR 56 mL/min/1.73 m^2^) in elderly patients versus 8.5% (average eGFR 79 mL/min/1.73 m^2^) in the non-elderly. When comparing eGFR before surgery and at discharge, an increase was seen: 13.76% in the elderly and 7.65% in the non-elderly patients, which was not significantly different. However, both increases were statistically significant from the baseline eGFR with a *p*-value of <0.001.

### 3.4. Long-Term Survival

Figure 2 shows the Kaplan–Meier curve of the long-term survival rate in elderly and non-elderly patients compared to the age-matched general Dutch population. Mean follow-up time was 6.2 ± 3.5 years. Long-term survival differed significantly between the elderly and non-elderly group with a *p*-value of <0.001. The five-year survival rate was 81.4% in the elderly and 93.9% in the non-elderly (*p* < 0.001) versus 86% and 98% in the age-matched general Dutch population. Moreover, survival in both study groups seemed lower than in the age-matched general Dutch population. Univariable analysis showed a history of hypertension (HR 2.64, 95% CI 1.21–5.78, *p* = 0.015) and diabetes mellitus (HR 0.24, 95% CI 0.06–0.97, *p* = 0.045) to be significantly associated with higher long-term mortality in the elderly, as well as in the non-elderly. In the non-elderly, male sex was significantly associated with a lower risk of long-term mortality, whereas in elderly patients this association was not found, as is shown in the Appendix A.

Figure 3 shows the multivariable analysis for the total population, and also for the elderly and non-elderly. Chronic kidney disease showed borderline significance with higher long-term mortality in the elderly (*p* = 0.053). In the non-elderly, higher age was associated with higher long-term mortality. Additionally, receiving Valve Sparing Aortic Root Replacement (VSARR) was associated with lower long-term mortality (the partial Yacoub technique was included in this multivariable analysis).

## 4. Discussion

In this large retrospective cohort study, elective AA surgery was compared in elderly versus non-elderly patients. Elderly patients, and elderly females in particular, had a significantly higher aneurysm diameter at the time of surgery. When it comes to surgical characteristics, SCAR, valve sparing procedures and surgery with concomitant procedures were performed more often in elderly patients, resulting in more complex procedures. Postoperative morbidity in the elderly was more prevalent, mostly due to supraventricular arrhythmias, delirium, and infections. Nevertheless, short-term mortality was not significantly different and was still relatively low (3%). Long-term mortality was higher in elderly patients, but seems to have a similar effect on life expectancy as non-elderly patients. No significant differences were found between elderly male and female patients.

### 4.1. Patient Characteristics

Previous studies showed that normal aortic diameters increase with age [13,14]. In our study, the elderly had a significantly higher absolute and indexed maximal aortic diameter at the time of AA surgery compared to the non-elderly, with a median absolute diameter of 57 mm, which is above the current threshold for elective surgery of 55 mm [15]. This suggests that elderly patients received surgery at a later stage in the disease process. This difference between elderly and non-elderly patients seems that it can be largely attributed to the fact that the elderly group contains more females (51.7% in elderly vs. 29.7% in non-elderly), since mean AA diameter before surgery was especially large in elderly females (61.2 mm). This is a very notable finding, since it was significantly larger compared to elderly males, which could suggest that surgery is performed at a later stage in elderly females. This might be because aneurysms in elderly patients are more often a coincidental finding, whereas in younger patients it is more in the context of (familial/known) aortopathy which is under regular follow-up care and when surgery is scheduled immediately upon reaching the operational limit. Despite the larger diameter, early outcomes in elderly males and elderly females were comparable, with a short-term mortality of 3.6% in elderly males and 2.5% in elderly females (*p* = 0.642). Furthermore, long-term mortality was not significantly associated with female sex in the elderly group. Therefore, it seems that elderly female patients did not suffer worse outcomes after surgery, despite being operated at a later stage in their disease process. When comparing risk factors, elderly males more often had prior cardiac or aortic surgery and more often had a BAV (26.8% vs. 10.9%, *p* = 0.002). It is well-known that BAV is more common in males. The latter might partially explain the male–female difference, since in patients with BAV, aortic aneurysm might be diagnosed earlier than in patients without BAV, due to family screening or heart murmurs. Furthermore, patients with BAV and additional risk factors such as persistent hypertension might receive surgery at lower diameters, according to current guidelines [15]. Nevertheless, elderly females received surgery far above the current threshold for elective aortic surgery, although our results do not provide any reason to refrain from performing surgery in elderly females. Perhaps patient preference could have played a role in this as well. More research is needed to explore this finding and the factors underlying this remarkable male–female difference.

### 4.2. Operative Characteristics

Our analysis showed that SCAR was performed significantly more often in the elderly. This seems logical since 89% of the elderly had a maximal aortic diameter located at the ascending aorta, whereas the non-elderly more often had a dilated root (9.6% vs. 24.1%, *p* < 0.001). This difference might be partially attributable to the higher number of patients with HTAD in the non-elderly group, who more often have aneurysms located at the root [16]. If valve surgery was performed, elderly patients were operated on with biological prostheses significantly more often than the non-elderly, as is recommended by international guidelines [17].

Furthermore, the analysis showed that receiving VSARR was associated with lower long-term mortality in non-elderly patients (HR 0.30, 95% CI 0.10–0.66), *p* = 0.044). The mean age of the non-elderly patients receiving VSARR in our analysis was 46.5 years. A possible explanation for this phenomenon is that aortic valve lesions are less complex at a younger age compared to an older age, and this could result in lower long-term mortality rates. Our results are comparable with an earlier study which reported that there was an improved midterm survival rate among adult patients undergoing VSARR [18].

We hypothesized that the operation itself might take a longer amount of time in the elderly compared to the non-elderly, especially, since more concomitant procedures were performed. However, this was not reflected in our results. This finding is in line with previous studies which evenly reported no significant differences in perfusion time, aortic cross-clamp time, and ACP time between elderly and non-elderly patients [7,10]. Various studies analyzed all these intraoperative times, though their findings were not consistent [7,8,10,19]. This difference can be explained by the fact that the age limit was different and that the inclusion and exclusion criteria were inconsistent in these studies (with regard to e.g., descending aorta). Since we included only patients who actually underwent aortic surgery, relatively more elderly patients might have been selected with better performance states and lower pre-operative burdens. Another important factor that has to be accounted for is that there were missing data in intraoperative times and therefore this finding has to be interpreted with caution.

### 4.3. Short-Term Postoperative Outcomes

Our study showed that the elderly needed longer in-hospital and ICU admission days, which corresponds with previous reports [9,10,20]. In particular, ≥20 days of hospitalization occurred more often in the elderly (*p* = 0.014). This was at least partially due to more reoperations for mediastinitis. Furthermore, elderly suffered more minor postoperative complications, especially supraventricular arrhythmias, delirium, and infections, which are known to be more common in the elderly [21,22,23].

More importantly, our in-hospital mortality rate after elective aortic aneurysm surgery was not significantly higher in the elderly. There is a disagreement in the literature regarding postoperative mortality after elective aortic aneurysm surgery. Some studies found higher short-term mortality in the elderly [7,10,20], while other studies found the postoperative mortality outcomes to be similar between the two groups [7,9]. Presumably this is explained by the fact that in the studies of Peters et al. and Guo et al., the elderly were aged 75 or 80 years and older instead of 70 years or older [10,20]. However, our sub-analysis of the elderly aged 75 years or older did not show a significant difference in short-term mortality either. This difference might be due to the fact that elderly patients in the studies of Peters et al. and Guo et al. had more cardiovascular risk factors, such as coronary artery disease, previous cardiac surgery, higher mean age, and higher mean maximal aortic diameter than our patient population [10,20].

Various factors can influence renal function during cardiopulmonary bypass (CPB), such as inflammatory responses, hypothermia, changes in hemodynamics, and surgical stress [24]. Figure 1 shows that estimated glomerular filtration rate (eGFR) decreased postoperatively, but the eGFR measured at discharge was ultimately higher than the last measured eGFR before surgery. It has been reported that cardiac surgery with CPB does not necessarily have to lead to a decrease in renal function in patients with preoperative mild renal dysfunction [24]. In fact, our study showed the same phenomenon: postoperatively, there was a significant increase in eGFR compared to the eGFR before surgery (*p* = 0.005). A possible explanation could be the small changes in hemodynamic variables [25] or the effect of the artificial kidney in the CPB. Another theory for this phenomenon is that the postoperative eGFR was increased due to medication. There have been frequent studies on medication that should offer renal protection. Unfortunately, none of them showed a decrease in renal damage [25]. Since the increase in eGFR also occurs in other studies, it is unlikely that this finding is a coincidence and further research is clearly warranted.

### 4.4. Long-Term Survival

As expected, long-term survival differed significantly between the elderly and non-elderly groups. Compared to the general Dutch population, the survival of both elderly and non-elderly patients seems lower. Visually, this difference seems more pronounced in the elderly group, which suggests a greater impact of elective aortic surgery on long-term survival in the elderly. However, this could not be statistically tested in this study.

In the non-elderly, higher age was associated with higher long-term mortality, whereas this was not the case in the elderly group. Perhaps the patients in the non-elderly group had more variability in age in the presence of other risk factors and comorbidities compared to the patients in the elderly group. The difference in mortality risk between younger adults and adults approaching 70 might therefore be larger than the difference within the elderly group, which leads to our finding of age being a more important risk factor within the non-elderly group. Furthermore, in non-elderly patients, female sex was associated with higher long-term mortality in univariable analyses. In multivariable analysis, sex was not significantly associated with long-term mortality. Therefore, this finding might be attributable to confounding, warranting attention in future studies.

### 4.5. Study Limitations

This study has several limitations. First, this is a retrospective cohort study, and for this reason, there were missing data. Second, as stated before, only patients who underwent surgery were included and the study did not include patients who were not operated on (for example due to their comorbidity burden) which will have caused inclusion bias. Third, the impact of aortic surgery on quality of life was not determined in our study. Since this is an important factor especially in the elderly, we feel this is an important topic to incorporate in future research.

Thus, this study brings valuable information about the risk of elective aortic surgery in elderly patients, with a sizeable cohort operated on over 10 years. Some next steps in this study’s exploration are related to the quality of life after an elective ascending aortic aneurysm surgery. Did the clinical condition of the elderly patients improve (or decline) after surgery and are they satisfied with the postoperative results? Or did they regret the choice they had made afterwards? This also brings useful information for the elderly who are considering invasive surgery. Besides, further exploration of the male–female difference in the elderly is necessary, since the results of our study suggests there may be differences.

## 5. Conclusions

This study showed that elective ascending aorta surgery is performed at a larger aorta diameter in the elderly, especially in elderly females who had very large pre-operative aortic diameters. Despite being operated on at a later stage of disease, postoperative mortality and major morbidity after elective aortic aneurysm surgery were not different in the elderly, nor in elderly females, compared to the non-elderly. Therefore, reluctance towards performing elective aortic surgery in selected elderly patients is less necessary. Further exploration of the male–female difference in elderly patients is warranted.

## Figures and Tables

**Figure 1 jcm-12-02015-f001:**
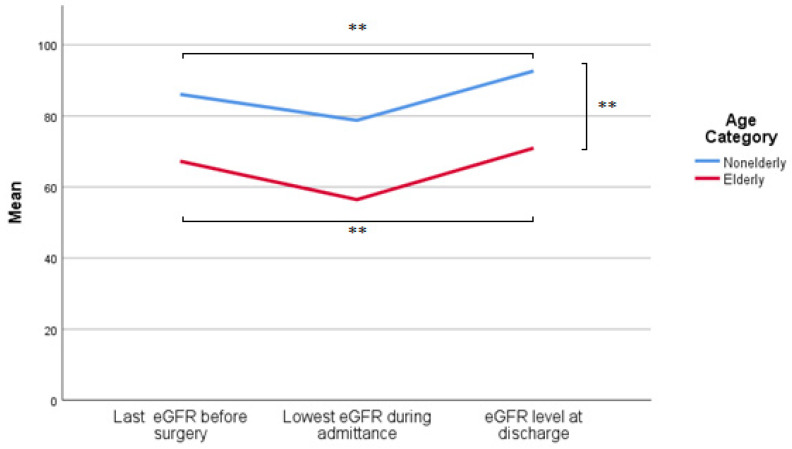
The Course of eGFR Level from Admission to Discharge. ** Significant at the *p* < 0.001 level.

**Figure 2 jcm-12-02015-f002:**
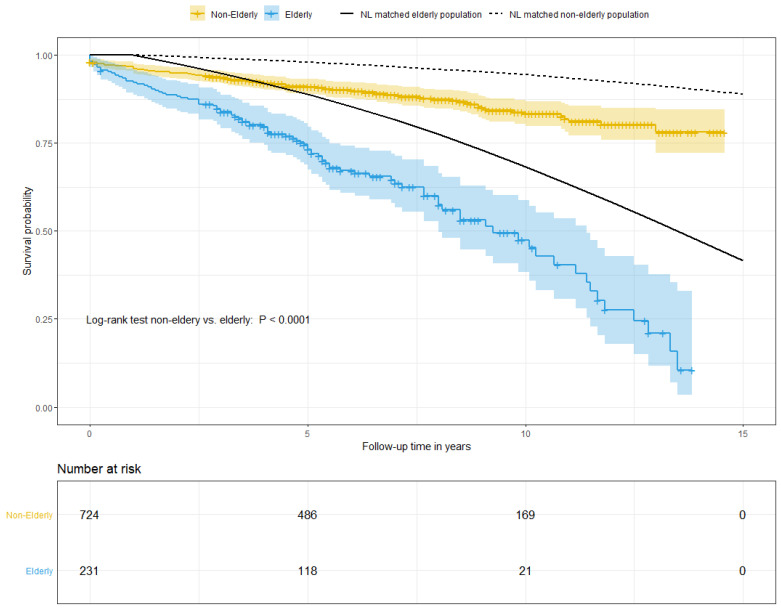
Kaplan–Meier Survival Analysis. Kaplan–Meier survival analysis for the elderly (blue) versus the non-elderly (yellow).

**Figure 3 jcm-12-02015-f003:**
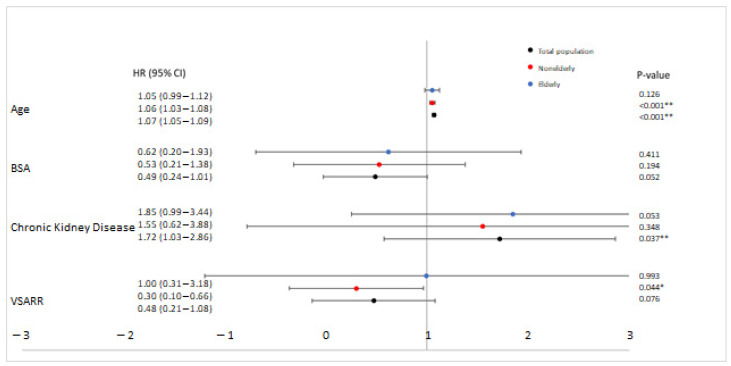
Forrest Plot of the Multivariable Analysis. Data are presented as Hazard Ratios (HR) with 95% Confidence Intervals (CI). VSARR: Valve Sparing Aortic Root Replacement (including partial Yacoub technique in this analysis); BSA: Body Surface Area. * Significant at the 0.05 level. ** Significant at the 0.01 level.

**Table 1 jcm-12-02015-t001:** Patient Characteristics.

	Total (*n* = 955)	Non-Elderly (*n* = 724)	Elderly (*n* = 231)	*p*-Value	Missings (NE/E) ^1^
Age (years)	61.0 (50.0–69.0)	55.0 (45.0–63.0)	74.0 (72.0–77.0)	<0.0001 **	0/955
Sex (% male)	621 (65.0)	509 (70.3)	112 (48.3)	<0.001 **	0/955
BSA	2.02 ± 0.22	2.05 ± 0.23	1.95 ± 0.17	<0.001 **	(1/724)/(0/231)
History of Hypertension	570 (59.7)	384 (54.4)	186 (81.6)	<0.001 **	(18/724)/(3/231)
History of Dyslipidemia	291 (30.5)	200 (28.5)	91 (40.3)	0.001 **	(22/724)/(5/231)
History of Diabetes Mellitus	62 (6.5)	41 (5.7)	21 (9.3)	0.067	(10/724)/(4/231)
History of CVA/TIA	110 (11.5)	74 (10.4)	36 (15.7)	0.034 *	(12/724)/(1/231)
History of COPD	90 (9.4)	60 (8.4)	30 (13.2)	0.039 *	(13/724)/(3/231)
History of Smoking					(302/724)/(87/231)
Never	182 (19.1)	134 (31.8)	48 (33.3)	0.76	
Currently	149 (15.6)	124 (29.4)	25 (17.4)	0.004 **	
In past	235 (24.6)	164 (38.9)	71 (49.3)	0.031 *	
History of Chronic Kidney Disease	48 (5.0)	25 (3.5)	23 (10.1)	<0.001 **	(10/724)/(3/231)
eGFR (mL/min)				<0.001 **	(0/724)/(1/231)
<60	121 (12.7)	56 (7.7)	65 (28.3)		
≥60	833 (87.2)	668 (92.3)	165 (71.7)		
History of Myocardial Infarction	50 (5.2)	37 (5.1)	13 (5.8)	0.73	(2/724)/(6/231)
Family History of Aortic Pathology	134 (14.0)	120 (42.0)	14 (19.7)	0.001 **	(438/724)/(161/231)
Diagnosis of HTAD Prior to Surgery	129 (13.5)	123 (17.0)	6 (2.6)	<0.001 **	(504/724)/(216/231)
Marfan Syndrome	48 (5.0)	47 (6.5)	1 (0.4)	<0.001 **	
Loeys–Dietz Syndrome	5 (0.5)	5 (0.7)	0 (0.0)	0.34	
SMAD3 Mutation	10 (1.0)	10 (1.4)	0 (0.0)	0.13	
Turner Syndrome	8 (0.8)	8 (1.1)	0 (0.0)	0.21	
Suspected	42 (4.4)	39 (5.4)	3 (1.3)	0.005 **	
Other	16 (1.7)	14 (1.9)	2 (0.9)	0.38	
Prior Cardiac Surgery	97 (10.2)	81 (11.2)	16 (6.9)	0.062	0/955
Prior Aortic Surgery	85 (8.9)	73 (10.1)	12 (5.2)	0.024 *	(1/724)/(1/231)
Aortic Valve					
Stenosis	221 (23.1)	178 (26.2)	43 (20.6)	0.099	(45/724)/(23/231)
Insufficiency	397 (41.6)	289 (42.2)	108 (50.7)	0.033 *	(39/725)/(19/231)
NYHA Classification					(64/724)/(23/231)
Class I	500 (52.4)	402 (60.9)	98 (47.1)	<0.001 **	
Class II	224 (23.5)	159 (24.1)	65 (31.3)	0.040 *	
Class III	135 (14.1)	91 (13.8)	44 (21.2)	0.010 *	
Class IV	9 (0.9)	8 (1.2)	1 (0.5)	0.695	
CCS Classification					(87/724)/(35/231)
Class I	711 (74.5)	551 (86.5)	160 (81.6)	0.092	
Class II	79 (8.3)	53 (8.3)	26 (13.3)	0.039 *	
Class III	36 (3.8)	27 (4.2)	9 (4.6)	0.84	
Class IV	7 (0.7)	6 (0.9)	1 (0.5)	1.00	
LVEF					(57/724)/(14/231)
Good (>55%)	680 (71.2)	524 (78.6)	156 (71.9)	0.043 *	
Reduced (44–55%)	112 (11.7)	80 (12.0)	32 (14.7)	0.29	
Moderate (30–45%)	85 (8.9)	60 (9.0)	25 (11.5)	0.27	
Poor (<30%)	7 (0.7)	3 (0.5)	4 (1.8)	0.067	
Bicuspid Aortic Valve					0/955 (21/724)/(12/231) (21/724)/(12/231) (21/724)/(12/231) 0/955
Location of Maximal Aortic Diameter	375 (39.3)	332 (45.9)	43 (18.6)	<0.001 **	
Sinuses of Valsalva	191 (20.0)	170 (24.1)	21 (9.6)	<0.001 **	
Ascending Aorta	722 (75.6)	527 (74.8)	195 (89.0)	<0.001 **	
Aortic Arch	10 (1.0)	7 (1.0)	3 (1.4)	0.71	
Descending Aorta	1 (0.1)	1 (0.1)	0 (0.0)	1.00	
Maximal Absolute Aortic Diameter (mm)	54.0 (50.0–59.0)	53.0 (49.0–58.0)	57.0 (53.0–63.0)	<0.001 **	
Maximal Indexed Aortic Diameter (mm/m^2^)	26.7 (24.0–30.3)	25.9 (23.4–29.2)	30.3 (26.9–34.5)	<0.00 **	
Logistic EUROscore	9.0 (5.3–14.4)	7.0 (5.0–11.6)	17.4 (13.0–22.7)	<0.001 **	

Continuous data are presented as mean ± SD when the distribution is normal, or median (Interquartile Range, IQR) for variables without normal distributions. Categorical data are presented as frequencies (percentages). BSA: Body Surface Area, CVA/TIA: Cerebrovascular Accident/Transient Ischemic Attack, COPD: Chronic Obstructive Pulmonary Disease, GFR: Glomerular Filtration Rate, HTAD: Hereditary Thoracic Aortic Disease, NYHA: New York Heart Association, CCS: Canadian Cardiovascular Society, LCC: Left Coronary Cusp, LVEF: Left Ventricular Ejection Fraction, Logistic EUROscore: European System for Cardiac Operative Risk Evaluation, NCC: Non Coronary Cusp, RCC: Right Coronary Cusp. ^1^ NE/E: Non-elderly/Elderly. * Significant at the 0.05 level. ** Significant at the 0.01 level.

**Table 2 jcm-12-02015-t002:** Operative Characteristics.

	Total (*n* = 955)	Non-Elderly (*n* = 724)	Elderly (*n* = 231)	*p*-Value	Missings (NE/E) ^1^
Bentall Procedure	491 (51.4)	389 (53.7)	102 (44.0)	0.010 *	0/955
Mechanical	324 (33.9)	304 (42.0)	20 (8.6)	<0.001 **	
Biological	167 (17.5)	85 (11.7)	82 (35.3)	<0.001 **	
David technique	117 (12.3)	110 (15.2)	7 (3.0)	<0.001 **	0/955
(Partial) Yacoub technique	24 (2.5)	17 (2.3)	7 (3.0)	0.63	0/955
SCAR	390 (40.8)	264 (36.5)	126 (54.3)	<0.001 **	0/955
No AVR	219 (22.9)	152 (21.0)	67 (28.9)	0.013 *	
Mechanical AVR	53 (5.5)	52 (7.2)	1 (0.4)	<0.001 **	
Biological AVR ^a^	89 (9.3)	44 (6.1)	45 (19.4)	<0.001 **	
Valve repair	27 (2.8)	14 (1.9)	13 (5.6)	0.003 **	
Concomitant Procedures	514 (53.8)	359 (49.6)	155 (66.8)	<0.001 **	0/955
(Hemi-)arch	392 (41.0)	269 (37.2)	123 (53.0)	<0.001 **	
CABG	106 (11.1)	67 (9.3)	39 (16.8)	0.002 **	
Mitral valve surgery	32 (3.4)	20 (2.8)	12 (5.2)	0.076	
Other ^b^	59 (6.2)	44 (6.1)	15 (6.5)	0.83	
Perfusion Time (min)	163.0 (125.0–198.0)	163.0 (124.0–198.5)	162.0 (129.5–197.3)	0.69	(7/724)/(1/231)
Aortic Cross-Clamp Time (min)	107.0 (83.3–136.0)	108.0 (84.0–138.0)	103.0 (82.0–132.5)	0.12	(8/724)/(4/231)
DHCA ^c^	420 (44.0)	290 (40.1)	131 (56.5)	<0.001 **	
Circulatory Arrest Time (min)	19.0 (15.0–28.0)	18.0 (14.0–27.0)	20.0 (16.0–37.0)	0.021 **	(450/724)/(108/231)
Cerebral Protection ^c^	420 (44.0)	290 (40.1)	131 (56.5)	<0.001 **	(3/724)/(3/231)
Antegrade Unilateral	26 (2.7)	19 (2.6)	7 (3.0)	0.82	
Antegrade Bilateral	388 (40.6)	268 (37.0)	120 (51.7)	<0.001 **	
ACP time (min) ^c^	33.0 (20.0–68.0)	29.0 (18.0–69.0)	35.0 (25.8–63.5)	0.26	(199/724)/(80/231)

Continuous data are presented as mean ± SD when the distribution is normal, or median (Interquartile Range, IQR) for variables without normal distributions. Categorical data are presented as frequencies (percentages). SCAR: Supracoronary Aorta Replacement, AVR: Aortic Valve Replacement, CABG: Coronary Artery Bypass Grafting, DHCA: Deep Hypothermic Cardiac Arrest, ACP: Antegrade Cerebral Perfusion ^a^ All patients who had reduction ascending aortoplasty had undergone biological AVR as well, ^b^ Other performed concomitant procedures can be found in the appendix, ^c^ If the procedure was performed. ^1^ NE/E: Non-elderly/Elderly. * Significant at the 0.05 level. ** Significant at the 0.01 level.

**Table 3 jcm-12-02015-t003:** Short-term Outcomes.

	Total (*n* = 955)	Non-Elderly (*n* = 724)	Elderly (*n* = 231)	*p*-Value	Missings (NE/E) ^1^
In-hospital or 30-day Mortality	18 (1.9)	11 (1.5)	7 (3.0)	0.16	0/955
Cause of Mortality					0/955
Cardiac (incl. Tamponade)	8 (0.8)	6 (0.8)	2 (0.9)	1.00	
Bleeding	3 (0.3)	2 (0.3)	1 (0.4)	0.57	
Aortic Rupture	2 (0.2)	2 (0.3)	0 (0.0)	1.00	
Organ Failure	1 (0.1)	0 (0.0)	1 (0.4)	0.24	
Sepsis	2 (0.2)	0 (0.0)	2 (0.9)	0.06	
Other	1 (0.1)	0 (0.0)	1 (0.4)	0.24	
Number of Days the Patient Was Admitted ^a^					0/955
1–4	23 (2.4)	18 (2.5)	5 (2.2)	1.00	
5–9	590 (61.8)	460 (63.5)	130 (56.3)	0.048 *	
10–14	205 (21.5)	154 (21.3)	51 (22.1)	0.78	
15–19	66 (6.9)	47 (6.5)	19 (8.2)	0.37	
≥20	71 (7.4)	45 (6.2)	26 (11.3)	0.014 *	
Total ^#^	8.0 (7.0–11.0)	8.0 (7.0–11.0)	7.0 (9.0–13.0)	0.028 *	
Number of Days in ICU 1 After Surgery ^a^					(20/724)/(4/231)
1–4	827 (86.6)	641 (91.1)	186 (81.9)	<0.001 **	
5–9	71 (7.4)	41 (5.8)	30 (13.2)	<0.001 **	
10–14	18 (1.9)	11 (1.6)	7 (3.1)	0.17	
15–19	5 (0.5)	4 (0.6)	1 (0.4)	1.0	
≥20	10 (1.0)	7 (1.0)	3 (1.3)	0.71	
Total ^#^	2.0 (2.0–3.0)	2.0 (2.0–3.0)	2.0 (2.0–4.0)	<0.001 **	
Number of Days on Ventilation Support After Surgery ^a^					(28/724)/(9/231)
1	551 (57.7)	455 (65.3)	96 (43.2)	<0.001 **	
2	308 (32.3)	205 (29.4)	103 (46.4)	<0.001 **	
3	19 (2.0)	10 (1.4)	9 (4.1)	0.027 *	
4	10 (1.0)	6 (0.9)	4 (1.8)	0.27	
≥5	31 (3.2)	21 (3.0)	10 (4.5)	0.29	
Total ^#^	1.0 (1.0–2.0)	1.0 (1.0–2.0)	1.0 (1.0–2.0)	<0.001 **	
Total Reoperations	230 (24.1)	155 (21.4)	75 (32.3)	<0.001 **	(127/724)/(59/231)
Bleeding	157 (16.4)	112 (15.5)	45 (19.4)	0.16	
Tamponade	31 (3.2)	21 (2.9)	10 (4.3)	0.29	
Mediastinitis	16 (1.7)	5 (0.7)	11 (4.7)	<0.001 **	
Cardiac Ischemia	1 (0.1)	1 (0.1)	0 (0.0)	0.76	
Other Visceral Ischemia	2 (0.2)	0 (0.0)	2 (0.2)	0.059	
Structural Valve Deterioration	1 (0.1)	1 (0.1)	0 (0.0)	0.57	
Non-Structural Valve Deterior.	1 (0.1)	1 (0.1)	0 (0.0)	0.76	
Endocarditis	1 (0.1)	1 (0.1)	0 (0.0)	0.76	
Other	20 (2.1)	13 (1.8)	7 (3.0)	0.19	

Continuous data are presented as mean ± SD when the distribution is normal, or median (Interquartile Range, IQR) for variables without normal distributions. Categorical data are presented as frequencies (percentages). ^#^ Presented as median (Interquartile Range, IQR) of the total number of days in the whole, non-elderly and elderly population. ICU: Intensive Care Unit ^a^ Significant differences were mainly found in the outliers in the number of days in elderly patients. ^1^ NE/E: Non-elderly/Elderly. * Significant at the 0.05 level. ** Significant at the 0.01 level.

**Table 4 jcm-12-02015-t004:** In-hospital Postoperative Morbidity After Elective Aortic Aneurysm Surgery.

	Total (*n* = 955)	Non-Elderly (*n* = 724)	Elderly (*n* = 231)	*p*-Value	Missings (NE/E) ^1^
Tracheostoma Implantation	8 (0.8)	5 (0.7)	3 (1.3)	0.41	(1/724)/(0/231)
New Permanent Heart Rhythm Disturbances	245 (25.7)	154 (21.3)	91 (39.4)	<0.001 **	(3/724)/(0/231)
Supraventricular	211 (22.1)	125 (17.3)	86 (37.2)	<0.001 **	
Ventricular	6 (0.6)	6 (0.8)	0 (0.0)	0.35	
AV-block	27 (2.8)	22 (3.0)	5 (2.2)	0.65	
Pacemaker or ICD Implanted	32 (3.4)	26 (3.6)	6 (2.6)	0.54	0/955
Myocardial Infarction or Ischemia	18 (1.9)	13 (1.8)	5 (2.2)	0.78	0/955
Infective Endocarditis	4 (0.4)	3 (0.4)	1 (0.4)	1.00	(2/724)/(1/231)
Non-Structural Valve Dysfunction	2 (0.2)	2 (0.3)	0 (0.0)	1.00	0/955
CVA/TIA	43 (4.5)	33 (4.6)	10 (4.3)	0.89	(1/724)/(1/231)
New Recurrence Nerve Lesion	12 (1.3)	11 (1.5)	1 (0.4)	0.31	0/955
Diagnosis of Delirium	146 (15.3)	80 (11.0)	66 (28.6)	<0.001 **	0/955
Diagnosis of Infection	130 (13.6)	82 (11.3)	48 (20.8)	0.001 **	0/955
Diagnosis of Sepsis	11 (1.2)	7 (1.0)	4 (1.7)	0.31	0/955

Continuous data are presented as mean ± SD when the distribution is normal, or median (Interquartile Range, IQR) for variables without normal distribution. Categorical data are presented as frequencies (percentages). AV-block: Atrioventricular block, CVA/TIA: Cerebrovascular Accident/Transient Ischemic Attack, ICD: Implantable Cardioverter Defibrillator. ^1^ NE/E: Non-elderly/Elderly. ** Significant at the 0.01 level.

## Data Availability

Data will be shared on request to the corresponding author with permission of the Size Matters project research group.

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
