# Peer review of "Elective Ascending Aortic Aneurysm Surgery in the Elderly"

_jcm, 2023, doi:10.3390/jcm12052015_

Round 1
Reviewer 1 Report
Thank you for the opportunity to review this work. I commend the authors on the stellar write-up of this paper. I have a few comments and questions.
Do the authors have any surgeon level data that can be included as a potential confounder when adjusting during Cox regression?
Were there any significant differences in outcomes between elderly or non-elderly patients form the different centers included in this study?
Were there any disproportionate differences in the elderly and non-elderly patients included from each of the different centers?
Were the surgeons in each center all following the same standards of postoperative care, particularly the prevention and management of infectious complications in the elderly?
The article starts by emphasizing the lack of guidelines to treat this disease in the elderly. In the Discussion, the authors should discuss what ground they have covered in their article, and outline some next steps in this direction.
Author Response
We would like to thank the reviewer for reviewing the manuscript, and providing important comments. We have done our very best to carefully address all comments and we have revised the manuscript accordingly. In the document, we respond to each comment and indicate which changes were made.

Reviewer 2 Report
I’ve read with interest the manuscript titled: “Elective Ascending Aortic Aneurysm Surgery in the Elderly” by Memis and coworkers. The following are my comments:
- The study brings valuable information about the risk of elective surgery in elderly patients, with a sizeable cohort operated over 10 years
- The use of Logistic Euroscore is deprecated, as it overestimates early risk, especially in elderly subjects [1] and those undergoing ascending aortic surgery [2]. The use of the Euroscore II risk score would’ve been preferable- bit even this one would overestimate mortality in elderly patients. Your early mortality was 3.6% in elderly males and 2.5% in elderly females.
- From a statistical point of vue, propensity matching would’ve been the method of choice to equalise risk factors between elderly and non-elderly groups, especially since the former group has more cardiovascular risk factors, as well as more survival-limiting comorbidities
- Nevertheless, even with propensity scoring, as the mortality was very low, it would require a much larger sample to detect meaningful differences in matched patients.
- Normal aortic diameters increase with age (please see [3 and [4]). This could be referenced in your text
- This is a valuable study, that brings new insight on the early surgical risk of aortic surgery in elderly patients and of high interest to your readers
- Collart et al, Eur J Cardiothorac Surg. 2005 Feb;27(2):276-80. doi: 10.1016/j.ejcts.2004.10.041.
- Nishida et al, Interact Cardiovasc Thorac Surg2014 Apr;18(4):446-50. doi: 10.1093/icvts/ivt524. Epub 2013 Dec 23.
- Devereux et al, Am J Cardiol. 2012 Oct 15;110(8):1189-94. doi: 10.1016/j.amjcard.2012.05.063. Epub 2012 Jul 6.
- Wolak et al, JACC Cardiovasc Imaging. 2008 Mar;1(2):200-9. doi: 10.1016/j.jcmg.2007.11.005.
Author Response

(The authors gave the same response as above.)
